# Improving the Thermostability of *Thermomyces lanuginosus* Lipase by Restricting the Flexibility of N-Terminus and C-Terminus Simultaneously via the 25-Loop Substitutions

**DOI:** 10.3390/ijms242316562

**Published:** 2023-11-21

**Authors:** Xia Xiang, Enheng Zhu, Diao Xiong, Yin Wen, Yu Xing, Lirong Yue, Shuang He, Nanyu Han, Zunxi Huang

**Affiliations:** 1School of Life Sciences, Yunnan Normal University, Kunming 650500, China; 2Engineering Research Center of Sustainable and Utilization of Biomass Energy, Ministry of Education, Yunnan Normal University, Kunming 650500, China; 3Key Laboratory of Yunnan for Biomass Energy and Biotechnology of Environment, Yunnan Normal University, Kunming 650500, China; 4Key Laboratory of Enzyme Engineering, Yunnan Normal University, Kunming 650500, China

**Keywords:** lipase, thermostability, B-factor, site-directed mutagenesis, N-terminus, C-terminus

## Abstract

(1) Lipases are catalysts widely applied in industrial fields. To sustain the harsh treatments in industries, optimizing lipase activities and thermal stability is necessary to reduce production loss. (2) The thermostability of *Thermomyces lanuginosus* lipase (TLL) was evaluated via B-factor analysis and consensus-sequence substitutions. Five single-point variants (K24S, D27N, D27R, P29S, and A30P) with improved thermostability were constructed via site-directed mutagenesis. (3) The optimal reaction temperatures of all the five variants displayed 5 °C improvement compared with TLL. Four variants, except D27N, showed enhanced residual activities at 80 °C. The melting temperatures of three variants (D27R, P29S, and A30P) were significantly increased. The molecular dynamics simulations indicated that the 25-loop (residues 24–30) in the N-terminus of the five variants generated more hydrogen bonds with surrounding amino acids; hydrogen bond pair D254-I255 preserved in the C-terminus of the variants also contributes to the improved thermostability. Furthermore, the newly formed salt-bridge interaction (R27…E56) in D27R was identified as a crucial determinant for thermostability. (4) Our study discovered that substituting residues from the 25-loop will enhance the stability of the N-terminus and C-terminus simultaneously, restrict the most flexible regions of TLL, and result in improved thermostability.

## 1. Introduction

Lipases (EC 3.1.1.3), commonly known as triacylglycerol lipases, are remarkable lipolytic enzymes capable of catalyzing a wide range of reactions, including esterification, hydrolysis, ammonolysis, and transesterification, in both aqueous and non-aqueous media [1,2]. Due to their versatile catalytic activities, lipases are applied in various industries such as foods, surfactants, pharmaceuticals, biofuel industries, and detergent industries [3,4,5]. Comparing lipases extracted from plants and animals, microbial lipases have particular advantages in commercial applications with regard to their ease in genetic manipulation and cultivation to obtain high yields [5,6]. Considering the severe biological treatment during the industrial process, denaturation limits the industrial applications of lipases to a large extent, including microbial lipases [7,8]. Consequently, lipase products with high expression yields, viable catalytic efficiency, and superior thermal and pH stability are in urgent need.

Lipases are widely used in many industries. The baking process in the food industry, the pelleting process in feed production, and the laundering process in the detergent industry require lipases with greatly enhanced thermostability at elevated temperatures [9,10]. To improve lipase thermostability, both directed evolution and rational design have been extensively applied as alternatives to provide thermally stable lipases [11,12,13]. In comparison with directed evolution—which requires the construction of a randomly generated mutational library followed by the complicated process of selecting, evaluating, and verifying work—rational design is widely applied since it can tailor-design lipases with improved performance using less effort [14,15]. For instance, based on B-factor analysis, Xie et al. constructed a double mutant D223G/L278M of lipase B from *Candida antarctica* at the active site, enhancing its kinetic activity and thermostability concurrently with a 13-fold increment in a half-life at 48 °C [16]. After introducing a proline to the lipase from *Yarrowia lipolytica* guided by molecular dynamics (MD) simulations and structural analysis, the optimal temperature of mutant V213P displayed a 5 °C improvement compared with its wild type [17]. Wang et al. combined the Gibbs free energy calculation (∆∆G) and MD simulations to investigate the mechanism of improving the thermostability of *Rhizopus chinenis* lipase (R27RCL); they finally obtained four positive variants (S142A, D217V, Q239F, and S250Y) and a combined variant, M31 [18]. All the above examples demonstrate the effectiveness and reliability of rational design in improving lipase thermostability.

*Thermomyces lanuginosus* is a very common thermophilic fungus in nature. Lipases from *Thermomyces lanuginosus* (TLL) have displayed beneficial properties over other lipases such as efficient catalytic activity, rapid reaction speed, and high heat and pH stability [19,20]. TLL has been used as a commercial lipase preparation by Novozymes in a soluble form, Lipolase^®^, and in an immobilized form, Lipozyme TL IM^®^ [19]. However, the development of the enzyme industry is limited by the higher temperature requirement in practical applications. To prevent the denaturation of TLL, it is necessary to improve its thermostability. Comparing lipases from other thermophilic organisms, TLL showed a slightly lower optimal reaction temperature (40 °C) [21]. In contrast, the optimum temperatures of lipases from *Enterobacter* sp. Bn12 and *Rhodothermus marinus* were 60 °C and 70 °C, respectively [22,23]. Accordingly, there is still space for TLL to improve its thermostability to become a widely used commercial product.

To improve the thermostability of TLL, many efforts have been tested. Zhu et al. demonstrated that glycosylation can improve TLL thermostability by mutating the glycosylation site (Asn-33) identified via LCMS/MS, indicating that this post-translational modification is important for TLL thermostability [24]. Qu et al. combined MD simulation and in silico mutation prediction to provide TLL variants with improved thermostability, and finally obtained variant G91C with both improved residual activity and a 5 °C increment in the optimal temperature without losing its specific and catalytic activity [25]. Han et al. substituted a surface-charged residue, R209, in TLL and finally obtained three single variants (R209M, R209H, and R209I) with both an increased optimal temperature and melting temperature directed by the ∆∆G calculation [21]. Gupta et al. reported the binary immobilization of TLL using chitosan as the support and resulted in increased thermostability against the crude enzyme [26]. Furthermore, integrating TLL immobilization and protein engineering is an alternative path for improving the thermostability of TLL [27].

In this study, we intend to improve the thermostability of TLL by substituting highly flexible amino acids with rigid ones. Specifically, residues with unusual flexibility in TLL were first identified using B-factor analysis. Subsequently, multiple sequence alignment was performed to select conserved sequences for the later substitutions. Finally, variants with improved thermostability were obtained, and it was discovered that the increased thermostability was due to restricting the most flexible regions of TLL simultaneously. Our study discovered the key 25-loop residues, which influence the stability of both the N-terminus and C-terminus in TLL, for the first time; the positive variants may offer new insights for future studies.

## 2. Results

### 2.1. Key Residues Selected for Improving TLL Thermostability

The B-factor determined using X-ray diffraction is linearly related to the mean square displacement of an atom relative to its average position [28]. Therefore, the B-factor values from a crystal structure provide meaningful information on protein flexibility [29]. It is worth mentioning that protein flexibility, especially in the loop regions, tends to be larger in the solution than that estimated from a crystal structure due to the crystal-packing effects [30]. In this study, the B-factors of TLL were first extracted from its crystal structures (PDB ID: 4ZGB and 1DT3). After normalization, residues with unusual flexibility (24-KNNDAPAG-31) located around the 25-loop (residues from 24–30) TLL were identified in both crystal structures (Figure 1A,B). Subsequently, the conserved sequence was analyzed based on 143 sequences derived from thermophilic fungal lipases using the ConSurf Server [31]; the consensus residues from 24 to 31 (24-SNNNASPG-31) which occupy the largest proportion on these sites were thereby confirmed (Figure 1C). Compared with the consensus sequence, four residues on sites 24, 27, 29, and 30 were identified as different from those in TLL. In addition, we manually substituted the flexible residues to other amino acids using the “mutagenesis” program in PyMOL [32], and discovered that replacing residue D27 with arginine (R) may generate more interactions and result in improved thermostability. Therefore, five variants were constructed for further experimental validation—K24S, D27N, P29S, A30P, and D27R. The crystal structure of TLL including the N-terminal region (NTR), C-terminal region (CTR), and the 25-loop is illustrated in Figure 1B.

### 2.2. Construction and Characterization of TLL and the Five Single Variants

As shown in the SDS-PAGE result (Figure 2), the five purified variants displayed similar molecular weights as TLL (30.05 kDa). The specific activities of purified TLL and variants K24S, D27N, D27R, P29S, and A30P were 318.7, 432.8, 375.6, 493.6, 371.8, and 308.9 U/mg, respectively. All five variants displayed comparable or increased specific activities relative to TLL.

To evaluate the thermostability of the 25-loop variants, the optimum reaction temperature (T_opt_), melting temperature (T_m_), and residual activities, under elevated temperature, of the TLL and five variants were measured. TLL displayed its maximal activity at 40 °C; thus, its T_opt_ was 40 °C (Figure 3A). For the five variants, they all displayed maximal activities at higher temperature. The optimum reaction temperature of all the five variants was 45 °C (Figure 3A). The residual activities after incubation at 80 °C for 90 min were measured; four out of five variants displayed comparable or improved thermostability. There were 60.9%, 46%, 52%, 54%, and 65% of the initial activities left for variants K24S, D27N, D27R, P29S, and A30P after the 90 min incubation, respectively, whereas there was 41% of the initial activity left for TLL under the same condition (Figure 3B, Table 1). It is worth noting that although the optimal reaction temperature of D27N increased, D27N exhibited decreased residual activity compared to TLL when incubated at 80 °C for the initial 75 min (Figure 3B). 

The melting temperature (T_m_) was determined using differential scanning calorimetries (DSCs). The apparent T_m_ value of the TLL was 93.2 ± 0.4 °C. Compared with TLL, the melting temperatures of the three variants (D27R, P29S, and A30P) were significantly increased. The T_m_ values of D27R, P29S, and A30P were 94.2 ± 0.2 °C, 95.5 ± 0.2 °C, and 98.3 ± 0.4 °C, respectively. Among all the variants, A30P revealed the highest melting temperature, which was 5.1 °C higher than that of the wild type. Surprisingly, the T_m_ values of K24S and D27N decreased by 1.4 °C and 1.8 °C, respectively (Figure 4, Table 1). T_m_ is the temperature at which 50% of the proteins in the population are unfolded. The higher the T_m_, the more stable the protein. T_m_ values of K24S and D27N decreased, indicating that the stability of these two variants decreased compared with that of TLL.

To investigate the additive effect of the combined mutations, two quadruple variants were also constructed (K24S/D27R/P29S/A30P and K24S/D27N/P29S/A30P). However, the two quadruple variants only remain at 4.6% and 6.7% of specific activities relative to wild-type TLL, respectively. It is meaningless to evaluate the thermostability of these two quadruple variants for their low activities. Therefore, parameters regarding thermostability were not measured for these two quadruple variants.

### 2.3. Kinetic and Productivity Analysis of the TLL and Five Variants

The kinetic measurements were determined and the apparent *K*_m_ (Michaelis constant) for TLL was 0.22 mM (Table 2). All of the variants have a larger *K*_m_, indicating a decreased substrate affinity compared to TLL. The turnover number (*k*_cat_) values for the five variants increased, suggesting that the 25-loop variants can convert more substrates to products per active site per unit of time. Most of the time, *k*_cat_ just equals the rate constant (*k*_2_) in the product forming step. *K*_m_ is a parameter that integrates two steps (substrate binding and product forming). Therefore, if *K*_m_ and *k*_cat_ increase together, it means that the reaction rates (*k*_1_) in the substrate-binding step of five variants are much lower, suggesting worse binding of the substrate compared to TLL. Nevertheless, all five variants displayed increased catalytic efficiency for their larger *k*_cat_/*K*_m_ values than that of the TLL. In brief, although the substrate binding affinities of the 25-loop variants decreased, their catalytic efficiencies all improved in comparison with TLL.

Since kinetics are determined from initial reaction rates, enzyme catalytic effects that accumulate over a long reaction time are usually overlooked [33]. To evaluate enzyme utility and potential industrial value, productivity analysis was employed to assess the catalytic capacity of TLL and the five variants (Figure 5). It was observed that D27N, D27R, P29S, and A30P hydrolyzed more than 0.7 mM pNPP relative to the TLL when incubated at 45 °C for 20 min, and the productivities of these four variants are obviously superior to that of TLL. For K24S, it displayed less pNPP hydrolysis than TLL after 5 min of incubation, suggesting a decreased productivity relative to the TLL. The higher productivities of D27N, D27R, P29S, and A30P were due to their higher thermostability over the duration of the hydrolysis reaction, and these four variants possess reliable enzyme utility and potential industrial value.

### 2.4. Stability of the NTR for TLL and Five Variants

To explore the mechanism of the improved thermostability for the five variants, MD simulations were conducted for TLL and all the variants. After comparing the root-mean-square fluctuations (RMSF) between TLL and the five variants, it was discovered that residues in the 25-loop (residues 24~30) of NTR displayed higher RMSF values in TLL than those in the four variants (K24S, D27N, P29S, A30P) which were designed using conserved sequence substitution, indicating that rigidities in the 25-loop in these four variants increased compared with that of the wild type (Figure 6A). Since the 25-loop was located at the NTR of the TLL, it can be concluded that the mutational sites of these four variants enhance the rigidity of the NTR of TLL, which is regarded as the most flexible region of an enzyme.

For the only variant D27R, which was not designed from consensus-sequence substitution, the rigidity of its 25-loop was not as increased as other variants. To generate more electrostatic interactions, the negatively charged residue Asp was replaced with a positively charged residue Arg in the variant D27R. However, when the D27R was exposed to the solvent, the positively charged Arg may form more interactions with solvent molecules and lead to higher flexibility and reduced stability.

To figure out the reason behind the improved thermostability of the 25-loop variants, interactions that may reduce the flexibility of the 25-loop were monitored along the MD simulation trajectories. After analyzing the hydrogen bonds formed between the 25-loop and surrounding residues in TLL and the five variants, it was discovered that the 25-loop in the five variants generated more hydrogen bonds than that in TLL. Specifically, the average number of hydrogen bonds formed between the 25-loop and surrounding residues in K24S, D27N, D27R, P29S, and A30P were 5.0, 3.7, 5.0, 4.9, and 3.9, respectively, while this value was only 3.3 in TLL (Figure 6B).

### 2.5. Stability of the CTR for TLL and Five Variants

Since the NTR and CTR in TLL are located close to each other (Figure 1B), we monitored the dynamics of CTR in TLL and the five variants similar to the analysis performed on the NTR. It was discovered that the CTR residues in the four variants designed using conserved sequence substitution also displayed increased stability by analyzing their RMSF values. Similarly, the flexibility of residue 250–254 in the CTR of D27R was comparable with that of TLL due to their charged amino acid on site 27 (Figure 7A).

The sidechain carboxyl group of D254 and the mainchain amine group of I255 potentially make contact with each other in TLL and the variants. During the MD simulations, the hydrogen-bond-forming probabilities between D254 and I255 were 79%, 74%, 38%, 70%, and 65% in K24S, D27N, D27R, P29S, and A30P, respectively; meanwhile, this hydrogen bond pair was not well preserved in TLL (26%) and resulted in decreased rigidity (Figure 7B). It is known that NTR and CTR are two ends of a protein, and they are commonly regarded as the most flexible region. In this work, the mutational site in the 25-loop not only stabilizes the configuration of the NTR, but also reduces the flexibility of CTR, leading to an enhanced thermostability of the variants.

### 2.6. Additional Salt-Bridge Interaction Formed in D27R

D27R was a variant predicted with enhanced stability based on our manual substitution. Its optimal reaction temperature, melting temperature, residual activity, productivity, and catalytic efficiency were all improved relative to those of TLL. D27R was discovered with flexible NTR and CTR in the MD simulation, which may due to its electrostatic interaction with the surrounding solvent molecules. To explore the mechanism of the enhanced thermostability for D27R, interactions formed between D27R and surrounding residues were monitored. It is discovered that R27 and E56 formed an additional salt-bridge interaction in the simulation (Figure 8A). After calculating the minimal distance between the guanidine group of R27 and the carboxyl group of E56, the probability of forming a salt-bridge interaction between R27 and E56 was 67.1% when we set a distance cutoff 0.6 nm, while there was only repulsive force between the sidechain groups of D27 and E56 in TLL (Figure 8B). This newly formed salt-bridge interaction in D27R accounted for its improved thermostability. Our study suggests that manually substituting amino acid on one particular site is also an effective strategy to produce valuable variant for single-point substitution.

## 3. Discussion

Exploitation of a thermal-tolerant enzyme is valuable for meeting practical industrial requirements. This study successfully improved the thermostability of lipase from *Thermomyces lanuginosus* through B-factor analysis and consensus-sequence substitutions using site-directed mutagenesis. The MD simulation results indicated that substituting residues on the 25-loop of TLL can not only stabilize the rigidity of NTR, but also reduce the flexibility of CTR, which is the key reason for the improved thermostability of the variants designed using consensus-sequence substitution. For the manually generated variant D27R, the newly formed salt-bridge interaction accounts for its improved thermostability. However, decreased substrate binding affinities were also discovered in the variants. Therefore, several points are worth discussing.

Protein flexibility refers to the variability and elasticity of a macromolecular structure. The significant flexibility of protein will lead to unfolding and denaturation, and result in reduced enzymatic activity. To improve protein stability, especially at high temperatures, protein rigidity needs to be increased; however, this may lead to a negative impact on its catalytic efficiency [34,35]. Therefore, a balance between enzyme thermal stability and catalytic activity needs to be considered when designing mutations for improving enzyme thermostability. In this study, although the thermostability of the 25-loop variants increased, their substrate binding affinities decreased to a certain extent. The decreased substrate binding affinities may be due to its closeness to the lid domain of the lipase (8.5 Å). The lid domain of a lipase is important in lipase-catalyzed reactions—it will transform from a closed state to an open state when the substrate binds to the binding pocket [19,36]. It is supposed that the mutation on the 25-loop may influence the lid operation and result in decreased substrate binding affinity.

It is known that thermostability may result in an additive effect when combining different beneficial mutations together. To investigate the additive effect of the combined mutations, two quadruple variants were also constructed (K24S/D27R/P29S/A30P and K24S/D27N/P29S/A30P). However, the specific activities of the two quadruple variants are only 14.6 and 21.4 U/mg, respectively. It is meaningless to evaluate the thermostability of these two quadruple variants for their low activities. Therefore, the additive effect of thermostability was not observed in this study. One study indicated that the antagonistic effect may arise from close contact between mutational sites [37]. In our study, the mutational sites are all located closely at the 25-loop; thus, it is hard to observe the additive effect of the thermostability based on the above finding.

In this study, variant A30P displayed the highest melting temperature, which was 5.1 °C higher than that of the wild type. Furthermore, its optimal reaction temperature and its residual activity at an elevated temperature were superior to that of TLL. Proline differs from all other amino acids since its side chain curls back to the preceding peptide-bond nitrogen and forms a five-member pyrrolidine ring. This pyrrolidine ring imposes rigid constraints on the N-Cα rotation, so the conformation of the proline residue as well as its upstream residue in a peptide chain are limited. Therefore, when substituting an amino acid to a proline, its conformational freedom would be restricted and result in improved rigidity [38,39]. In this study, the substituted proline may decrease conformational entropy and result in increased rigidity, and the variant A30P will be a valuable industrial product for further development.

## 4. Materials and Methods

### 4.1. Materials

The site-directed mutagenesis kit and DMT chemically competent cells were purchased from Trans Gen (Beijing, China). The markers of DNA and proteins, and restriction endonucleases (*Eco*R I, *Not* I), were purchased from TaKaRa (Otsu, Japan). *Pichia pastoris* GS115 was purchased from Invitrogen (Shanghai, China). The mutagenic primers were synthesized by Shuoqing (Kunming, China). All other chemicals were commercially available and of analytical grade.

### 4.2. Gene Cloning and Site-Directed Mutagenesis

The variants of TLL were constructed via site-directed mutagenesis, and *TLL*-pPIC9K was used as the template to construct the sequences of five variants (*K24S*, *D27N*, *D27R*, *P29S*, and *A30P*). The mutations were introduced using the Fast MultiSite Mutagenesis System according to the manufacturer’s instructions. The forward and reverse primers for all the variants are listed in Table 3. The PCR procedure was set as 5 min at 94 °C, followed by 30 cycles of 30 s at 94 °C, 2 min at 55 °C, and 15 min at 72 °C. The purified PCR products were transformed into *E*. *coli* DH5α cells and verified via DNA sequencing (Sangon Biotech, Shanghai, China).

### 4.3. Protein Expression and Purification

Wild-type TLL-pPIC9K plasmids and their variants were extracted using a Plasmid Mini Kit I, linearized with *Sal* I, and then transformed into *Pichia pastoris* GS115 via electroporation individually. The transformants were selected on YEPD (yeast extract peptone glucose) plates containing 200 μg/mL of geneticin (G418). The selected clones were incubated in a BMGY liquid medium at 30 °C for 2 days. The thallus was transferred to a BMMY liquid medium with 2% anhydrous methanol at 30 °C for 2 days to induce the protein expression. All enzymes labeled with His-tag at the N-terminus were purified with the Ni-NTA agarose column using the 0–2 M imidazole solution. Enzyme concentrations were determined using the Bradford Protein Assay Kit. The protein content of TLL, K24S, D27N, D27R, P29S, and A30P was 3.67, 3.14, 3.19, 2.95,3.14, and 3.11 mg/mL, respectively.

### 4.4. Assessment of Lipase Activity

The lipase activity was determined using pNPP (4-nitrophenyl palmitate) as the substrate. One unit (IU) of lipase activity was defined as the number of enzymes releasing 1 μmol of p-nitrophenol per minute [40,41]. Thermostability was determined by measuring the residual enzyme activity of TLL and variants after incubation at 80 °C for 90 min under the optimal pH (9.0). The kinetic parameters were determined at 37 °C and pH 9.0 in Tris-HCl buffer (100 mM) at 12 concentrations of pNPP varied between 0.25 and 10.0 mM (0.25, 0.5, 1, 2, 3, 4, 5, 6, 7, 8, 9, 10 mM). Kinetic parameters *K*_m_, *V*_max_, and *k*_cat_ were calculated by fitting them to the Michaelis–Menten function using GraphPad Prism 8.4.3 (Pattern Software Inc., La Jolla, CA, USA). To ensure the reliability of the data, each experiment was replicated three times.

### 4.5. Assessment of Lipase Productivity

Hydrolysis of pNPP using TLL and the five variants was performed at 45 °C and pH 9.0. The substrate amount was 0.6 mL (1.39 g/L) and the enzyme loading was 1 mL which were mixed into the 8.4 mL Tris-HCl (100 mM) buffer to initiate the reaction. Samples were taken at different periods of incubation to monitor the time–course of lipase hydrolysis, and the reaction was terminated by adding SDS and Na_2_CO_3_. The pNPP content in the sample was assessed by measuring absorbance value at 405 nm. Comparing absorbance values at different time points allowed for the quantification of enzyme productivity [33]. The productivity experiments were repeated three times.

### 4.6. Prediction of the Mutagenesis Sites via B-Factor Comparison

To evaluate the flexibility of the protein, B-factors of the Cα atoms in TLL were extracted from the crystal structure file (PDB ID: 4ZGB and 1DT3) [42,43]. The B-factor values of each atom were normalized to have a distribution of zero mean and unit variance based on the following equation:$$B^{\prime }=\frac{B\;-<B>}{\sigma \left(B\right)}$$
where <B> is the average of all atoms, and σ(B) is the standard deviation of B-factor values. The above equation has been verified and applied by a previous study [44].

### 4.7. MD Simulation Details

The crystal structure of TLL (PDB ID: 4ZGB) was used as the wild-type structure. It was also used as the template to construct structures of the variants using the SWISS-MODEL server [45]. After 1000 steps of energy minimization, all the systems were equilibrated for 5 ns under NVT ensemble followed by another 5 ns equilibration under NPT ensemble, and 100 ns MD simulations were subsequently conducted for all the systems. All simulations were performed in an isothermal and isobaric ensemble (45 °C, 1 bar). Each system was solvated with TIP3P water in a cubic box [46]. The minimum distance between the protein and the edge of the box was set to 1.0 nm. The GROMACS program suite version 2022 and the Amber ff99SB force field were applied in the simulations [47]. The bond length constraints were applied to all bonds that contained hydrogen atoms based on the LINCS protocol [48]. Electrostatic interactions were treated with the Particle Mesh Ewald method with a cutoff of 1.0 nm with grid spacing for the FFT grid < 0.16 nm [49]. The root-mean-square fluctuation was calculated based on the entire residue using the program “gmx rmsf” in GROMACS. The program “gmx hbond” in GROMACS was applied to analyze the hydrogen bonds between all possible donors and acceptors [50]. Molecular images were made using the PyMOL software (version 1.3) [32]. The tpr files including the initial velocities for all the simulation systems were attached in the Appendix A for reproduction.

## 5. Conclusions

This study applied B-factor analysis and consensus-sequence substitutions to provide TLL variants with improved thermostability. It was discovered that mutations on the 25-loop can not only stabilize the salt-bridge interaction in the N-terminus, but also strengthen the hydrogen bonding network in the C-terminus of TLL. This is the first report of lipase variants which have enhanced thermostability restricting both NTR and CTR simultaneously. In this work, we discovered a dominant region, the 25-loop, at the N-terminus. The experimental validation and simulation results suggested that substitutions in the 25-loop will enhance TLL thermostability. Future studies could investigate, in depth, the NTR and CTR in the TLL.

## Figures and Tables

**Figure 1 ijms-24-16562-f001:**
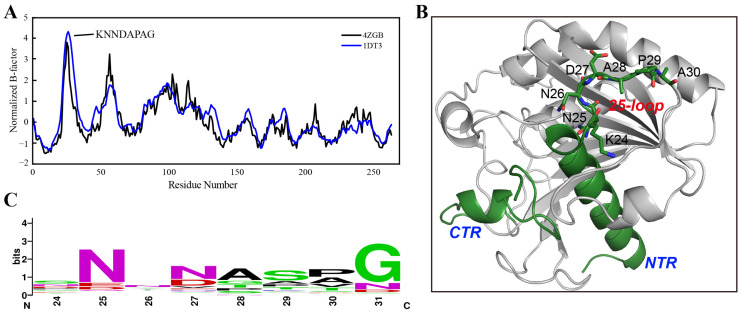
B-factor analysis and consensus-sequence identification. (**A**) Normalized B-factor values illustration based on an analysis of the crystal structure information of TLL (PDB ID: 4ZGB and 1DT3). (**B**) The N-terminal region (NTR), C-terminal region (CTR), the 25-loop, and residues within the 25-loop are illustrated in the TLL crystal structure (PDB ID: 4ZGB). (**C**) The consensus sequence was identified using the ConSurf Server based on 143 lipase sequences derived from thermophilic fungi.

**Figure 2 ijms-24-16562-f002:**
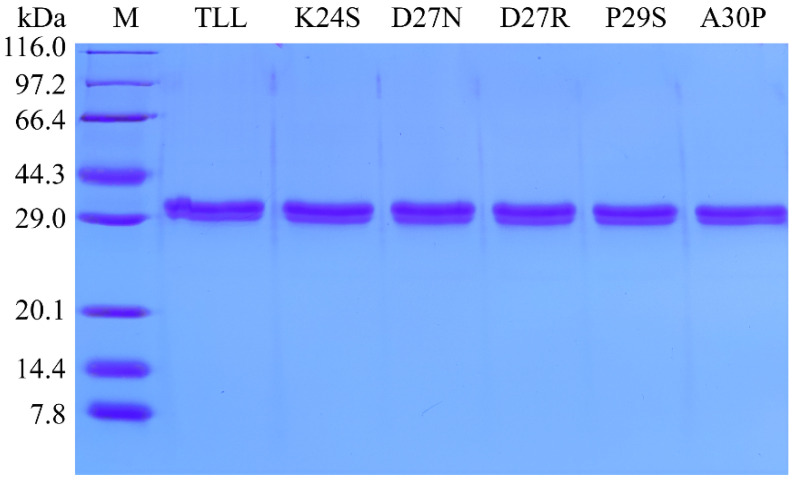
SDS-PAGE analysis of the purified enzymes of TLL and its variants.

**Figure 3 ijms-24-16562-f003:**
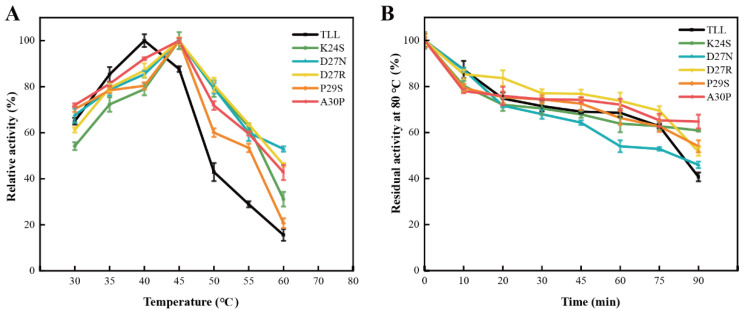
The optimal reaction temperatures and residual activities of TLL and five variants. (**A**) The optimal reaction temperatures of TLL and five variants. (**B**) The residual activities of TLL and five variants tested by incubating all lipases at 80 °C for 90 min. The data were measured three times at pH 9.0 in Tris-HCl buffer (100 mM) with pNPP (4-nitrophenyl palmitate) as the substrate. Error bars in the figure indicate the standard error of three repetitions.

**Figure 4 ijms-24-16562-f004:**
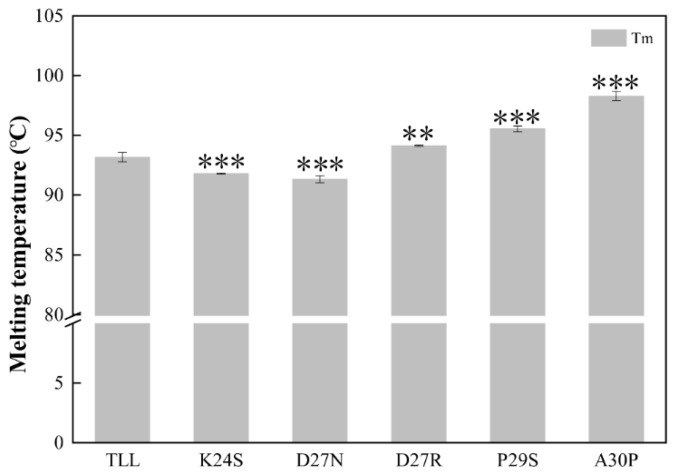
The melting temperatures (T_m_) of TLL and five variants. All the data were measured three times. The error bars in the figure indicate the standard error of three repetitions. The ANOVA results indicate the differences in bars; two and three stars between the variants and TLL are significant at the 0.01 and 0.001 levels, respectively.

**Figure 5 ijms-24-16562-f005:**
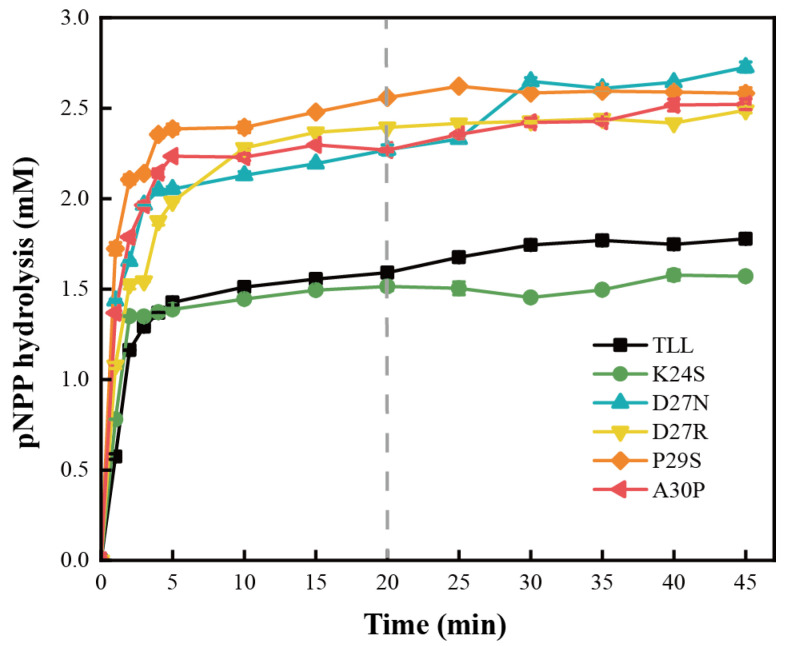
Productivity curves for TLL and the five variants. The productivities of TLL and the five variants tested by incubating the same amount of lipases at 45 °C for 45 min. The data were measured three times at pH 9.0 in Tris-HCl buffer (100 mM) with pNPP (4-nitrophenyl palmitate) as the substrate. Error bars in the figure indicate the standard error of three repetitions.

**Figure 6 ijms-24-16562-f006:**
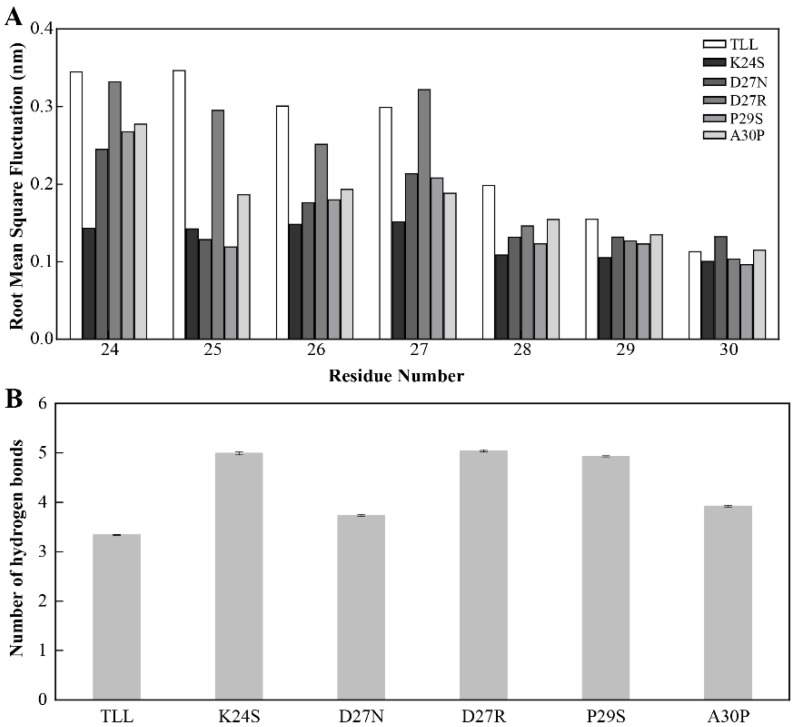
Stability of the NTR for TLL and the five variants. (**A**) The root-mean-square fluctuation (RMSF) of residues comprising the 25-loop (residues from 24 to 30) in TLL and the five variants. (**B**) Number of hydrogen bonds formed between the 25-loop and surrounding residues in TLL and five variants. The RMSF values in panel A were calculated based on the entire simulation trajectories. The error bars in panel B were calculated by randomly dividing the snapshots of the entire trajectories into five groups, and the standard errors were calculated based on these five groups.

**Figure 7 ijms-24-16562-f007:**
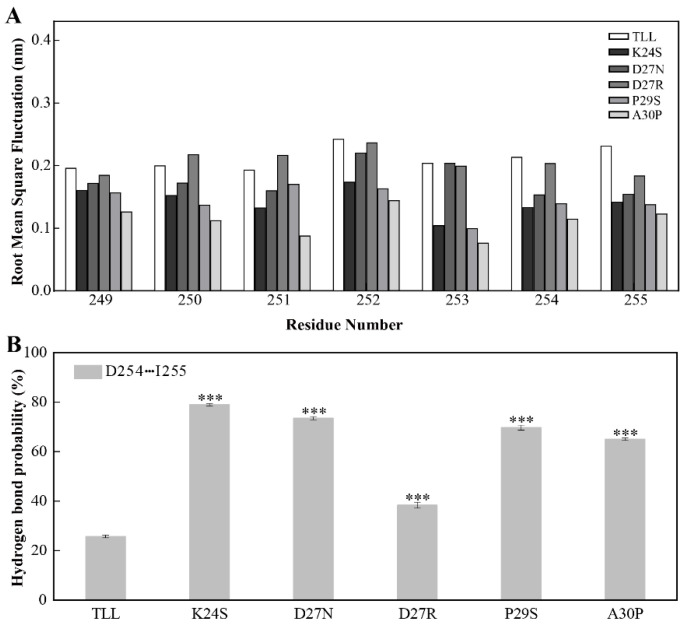
Stability of the CTR for TLL and the five variants. (**A**) The RMSF of residues comprising CTR (residues from 249 to 255) in TLL and the five variants. (**B**) Hydrogen-bond-forming probability of D254 and I255 in TLL and the five variants. The RMSF values in panel A were calculated based on the entire simulation trajectories. The error bars in panel B were calculated by randomly dividing the snapshots of the entire trajectories into five groups, and the standard errors were calculated based on these five groups. The ANOVA results in panel B indicate the differences in bars; three stars between the variants and TLL are significant at the 0.001 level.

**Figure 8 ijms-24-16562-f008:**
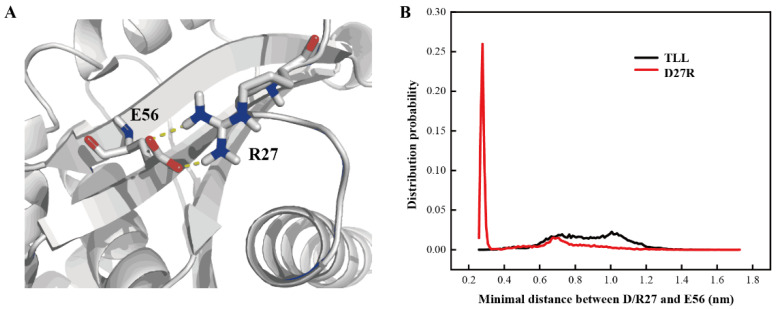
Additional salt-bridge interaction in D27R. (**A**) Salt-bridge interaction between R27 and E56 in representative structure. (**B**) Distance distribution between the guanidine group of R27 and the carboxyl group of E56 in D27R, as well as the carboxyl groups of D27 and E56 in TLL.

**Table 1 ijms-24-16562-t001:** Residual activities and melting temperatures of TLL and the five variants.

Enzymes	TLL	K24S	D27N	D27R	P29S	A30P
Residual activity (%)	41 ± 2	60.9 ± 0.5	46 ± 1	52 ± 2	54 ± 3	65 ± 3
T_m_ (°C)	93.2 ± 0.4	91.8 ± 0.1	91.4 ± 0.3	94.2 ± 0.2	95.5 ± 0.2	98.3 ± 0.4

**Table 2 ijms-24-16562-t002:** Kinetics of TLL and five variants.

Enzymes	*V*_max_ (µmol/min/mg)	*K*_m_ (mM)	*k*_cat_ (/s)	*k*_cat_*/K*_m_ (/s/mM)
TLL	402 ± 2	0.22 ± 0.05	594 ± 4	2722 ± 54
K24S	681 ± 11	0.31 ± 0.02	1379 ± 22	4485 ± 157
D27N	396 ± 2	0.22 ± 0.02	777 ± 5	3503 ± 31
D27R	828 ± 9	0.36 ± 0.01	1895 ± 21	5245 ± 118
P29S	668 ± 5	0.29 ± 0.03	1354 ± 9	4747 ± 28
A30P	503 ± 2	0.24 ± 0.02	1036 ± 4	4383 ± 42

**Table 3 ijms-24-16562-t003:** Oligonucleotide primers for the five variants.

Primers	Primer Sequences
*K24S*-forward	CATACTGCGGATCAAACAATGATG
*K24S*-reverse	TTGATCCGCAGTATGCGGC
*D27N*-forward	TGCGGAAAAAACAATAATGCCCCAGCT
*D27N*-reverse	CATTATTGTTTTTTCCGCAGTATGCGGCTG
*D27R*-forward	TGCGGAAAAAACAATCGTGCCCCAGCT
*D27R*-reverse	CACGATTGTTTTTTCCGCAGTATGCGGCTG
*P29S*-forward	AAACAATGATGCCTCAGCTGGTA
*P29S*-reverse	CTGAGGCATCATTGTTTTTTCCGC
*A30P*-forward	AACAATGATGCCCCACCTGGTACAAACA
*A30P*-reverse	CAGGTGGGGCATCATTGTTTTTTCCGCA

## Data Availability

Data supporting reported results can be found in the Appendix A.

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
