# Peer review of "Improving the Thermostability of Thermomyces lanuginosus Lipase by Restricting the Flexibility of N-Terminus and C-Terminus Simultaneously via the 25-Loop Substitutions"

_ijms, 2023, doi:10.3390/ijms242316562_

Round 1
Reviewer 1 Report (Previous Reviewer 2)
Comments and Suggestions for Authors
This is a resubmission of a manuscript that I had previously reviewed. The topic remains the same, Xiang et al. introduced mutations in a secondary structure of an enzyme called a 25-loop, which is named that after the number of one of the residues which creates that loop. After introducing mutations which should alter the thermostability of the enzyme and thus its flexibility, the authors have indeed reported that in their manuscript. They claim that introducing mutations increased the number of hydrogen bonds created with the N-terminal and C-terminal part of the enzyme, hence the stabilizing effect.
Indeed, the authors showed that the enzymes with introduced mutations are displaying mostly higher melting temperature as well as optimal reaction temperatures and residual activities (except for D27N variant). They also created the structures of mutated enzymes and ran MD simulations to investigate the origin of the increased stability of the enzymes. They showed that the mutant enzymes display lower flexibility and interact (via h-bonds) with the N- and C-terminal residues.
As I previously suggested, it would be interesting to analyse the dynamics and hydrogen bonds formation between the 25-loop and both N- and C-terminal regions of the quadruple mutants to investigate the reason of lower activity of the enzymes.
However, I understand that such an analysis may be outside of the scope of this manuscript. Thus, I will suggest to publish the manuscript, as I got no comments now. (I provided some comments previously and they were all addressed in this version of the manuscript). Good job!
Author Response
Please see the attachment.

Reviewer 2 Report (New Reviewer)
Comments and Suggestions for Authors
Science questions/improvements
Why were these 4 residues chosen? That was not made clear. I thought that 26 would be chosen since it was not highly conserved (like 24). A brief explanation for the rationale for the 4 that were chosen would be appropriate.
Paragraph lies 155-161
Why do you think the K24S and D27N decreased by those amounts? A small analysis right here would be helpful.
Section 2.3- lines 162-176
A few times it is mentioned that the Km goes up and that means lower affinity. It likely worth noting that Km has the rate constants for binding and conversion of substrate to product in it. Thus, if the Km is increasing but the kcat is also increasing, that means that binding is much worse than for the wild type. It would be interesting if you could test the actual binding constants to show how much worse they are at binding substrates/substrate analogs.
Lines 298-306: I was completely surprised that the combined mutants were made since there was no mention of them in the results. I would put a sentence in the results talking about them there. And then in the discussion it can be explored about how they are not additive and actually decrease stability.
Lines 311-315
I am not familiar with the “proline rule” and they way that it is stated makes no sense since the sentence does not describe what the upstream residues have to be/do. Please rewrite to be understandable.
Line 323- referring to the wild type plasmid- has that been published? Is there a discussion of it anywhere in the literature? Please reference if it has been. If not, please describe. If it is proprietary, indicate that.
A few grammar / sentence structure/ readability improvements:
Line 87- what is crude-free enzyme? Is that just pure protein? If so, change to “pure protein” but if it is something else, please explain what it is.
Line 105- should be B-factors (plural) since you are referring to multiple atoms in the protein.
Lines 140-144 and 155-161 Any time there are lists of numbers in the text, I find that having a table is easier to read. These two sets of data could be made into one small table that would make the text more readable.
Line 159-160 should make into two sentences. They should read: “…higher than that of the wild type. Surprisingly, the Tm values of K24S…”
Section 2.3- lines 162-176
I would suggest a big rearrangement of sentences here.
Starting at the end of line 167, I would suggest the following rewritten paragraph to make the results more clear. “The kinetic measurements were determined and the apparent Km (Michaelis constant) for TLL was 0.22 mM. All of the variants have a larger Km, indicating a decreased substrate affinity compared to TLL.
Line 184 I would remove the semi colon and write it this way instead “…45 oC for 20 minutes, suggesting that the productivities….”
Section 2.4- Please clarify that this section is looking at the NTR in the text at the start of the section. When I read section 2.5, it was confusing because it was so similar to 2.4, until I realized that the two sections were talking about NTR vs. CTR.
Line 231 change closely to close to each other.
Line 232, remove the words “as well” and add “, like the analysis performed on the NTR”
Line 256- with the word “Although” at the start of the sentence, it is a fragment sentence. Remove the word although, and then capitalize Is and the sentence is fine.
Comments on the Quality of English LanguageA few English improvements:
Several times, there are words that are not inaccurate, but give an outsized weight to the term appear. Usually a different word could be used.
Line 36 “Immense fields” should be changed to something like “large industries”
Line 47 “eminent thermostability” should be changed to “lipases with greatly enhanced thermostability”
Line 64 “is a kind of wide-spread thermophilic fungi” should be changed to “is a very common thermophilic fungi”
Line 66 “excellent heat” should be changed to “high heat”
Line 75 “for being a qualified commercial product” should be changed to “to become a widely used commercial product”.
Line 90 and Line 106 “remarkable flexibility” should be changed to “unusual flexibility”
Author Response
Please see the attachment.

This manuscript is a resubmission of an earlier submission. The following is a list of the peer review reports and author responses from that submission.
Round 1
Reviewer 1 Report
Comments and Suggestions for Authors
Xiang et al present a mutational study of the T. lanuginosus lipase (TLL). The mutations are rationally designed, based on bioinformatics analysis. The manuscript claims that some of these mutants show significant improvements in protein stability. However, the data presented are not convincing, with clear issues in key parts of the manuscript. My assessment of the data is that they show largely negative results with no significant improvement in enzyme activity. Given the reasonable rationale for the design, this is an interesting result. However, the manuscript should reflect the outcome and accept that this is an incremental study demonstrating useful but largely negative results.
Major issues:
The text revolving around figure 3 claims that there is a 5 C increase in optimal temperature for the variants. Figure 3A has too many overlapping points to make a judgment on this claim, and the differences are all between points showing over 95% activity. It is not clear at this level that any differences are significant. Furthermore, when considering the overall thermostability, it appears that in fact the wild-type TLL retains activity better than the mutants, with loss of activity occurring 5 C later than most mutants. Given the data presented, the claim that these mutants increase the optimum activity temperature cannot be supported. The authors should remove this claim from their manuscript and more accurately describe the results obtained.
Figures 6 and 7: the manuscript does not discuss how hydrogen bonds are assessed. This is particularly the case in figure 7: whilst the D27N hydrogen bond is convincing, those in D27R and P29S are dubious and those in K24S and A30P are not hydrogen bonds, as aspartic acid side chain oxygens only form hydrogen bonds in the plane of the acid group (as the oxygens and linking carbons are sp2 bonded). This suggests that the assessment of hydrogen bonds has been taken on an overly simplistic basis, and that consequently the analysis shown in Figures 6 and 7 is flawed and should be repeated with a more robust assessment of hydrogen bonding.
Figure 8A: this figure claims that it shows increased flexibility for the D27N variant. Given that only a single trajectory is shown, and that this trajectory stabilises with a similar RMSD to those of TLL and D27R, it is not clear that this is not merely a feature of the individual trajectory. Given the data shown, the statement made cannot be supported.
Other issues:
The introduction is substantially under-referenced. Some parts of the introduction have no references. Examples are lines 40-42, 48-53. The relevant literature should be cited to a much greater extent.
Lines 71-72: there is a substantial literature on the industrial use of lipase from T. lanuginosus. The authors’ statement that the thermostability needs to be increased to be a commercial product is clearly too strong. This statement should be amended and this literature on the use of the lipase should be cited.
Line 75: What is the “25 loop”? An image of the protein, clearly indicating what is meant, should be included.
Lines 84-87: it is true that B-factors are correlated with conformational flexibility in proteins. However, this statement should be nuanced with the fact that flexibility will be affected by crystal effects that are not related to flexibility in solution (a good discussion is found in doi:10.1016/j.biotechadv.2013.10.012).
Lines 91-92: the figure shows that 24-SNNNASPG-31 are the consensus residues. This does not necessarily mean that they are beneficial in the context of the TLL protein that will have other sequence variation that may make them non-beneficial.
Lines 94-95: the authors should explain how it was discovered that the D27R substitution may improve thermostability as this is not implied by the consensus sequence.
Figure 3: the number of repeats for each measurement should be indicated in the legend. The legend should include description of salient features of the reaction conditions (e.g. pH, substrate).
Lines 125-129 and elsewhere (especially table 1): determined values should be reported with their errors and should be reported to an appropriate precision. In most cases there is only sufficient data collected to give one significant figure of precision in the standard error; the values determined should be reported to similar accuracy. This is correctly done in Table 1 for the determined KM values but not for other values.
Figure 3, 4, 6, and 7 legends: please indicate what the error bars represent, and the statistical tests used.
Lines 142-143: although the K24S and D27R mutants show increased determined values of kcat/KM, these increases are not statistically significant (especially not D27R). The authors should remove the claim that these mutants show increased catalytic efficiency as this cannot be supported by the data presented to a reasonable level of significance.
The sentences in lines 154-157 should be moved to the figure legend.
Figures 5-8: please indicate the software used to make molecular images.
The paragraph in lines 165-176 is challenging to understand and could be ordered better. Specifically, it would be better to introduce first that in three mutants a main chain hydrogen bond is commonly formed whilst in two others, a salt bridge is formed; and that both are rare in the wild-type TLL.
Figure 6B: in the A30P panel, the conformation of K24 shown appears highly unlikely and not reflective of lysine conformations found in experimental protein structures. This does not give confidence that the molecular dynamics is proceeding effectively. Perhaps a more convincing image could be selected?
Lines 198-199: the references quoted do not support the statement made. Reference 18 concerns the deliberate generation of salt bridges, not “surface charge distribution”; reference 19 (the PI’s previous work) showed some minor improvements in activity in some cases of mutations of charged amino acids, but not in a systematic manner or more than might be expected by mutating other classes of amino acids. Whilst figure 8 shows a change in surface electrostatics caused by mutating a negatively charged residue to a neutral or positively charged one, this does not seem particularly meaningful.
Lines 256-259: these sentences are extremely hard to read and should be re-written to be clearer.
Lines 253-260: the claims that TLL A30P represents a significant improvement on the wild-type need to be more nuanced. The data presented do show improved thermostability and lifetime after incubation at 80 C. However, the optimal temperature is not significantly higher, and activity at higher temperatures is lower so this longer lifetime is of limited significance.
Section 4.3: the description of the expression and purification lacks detail. Details of the expression volumes, cell lysis, purification column, buffers used, and protein assay kit should be provided.
Lines 294-295: please indicate the number of different concentrations tested and the number of replicates of each concentration for the kinetic parameters.
Lines 302-303: the equation used is clearly wrong. By taking an absolute, it will show residues with very low B-factors similarly to those with B-factors the same distance from the mean. This means that it will not produce a distribution of zero mean as the text suggests. An example of this error is that the last 10 or so amino acids appear to have a moderately high B-factor. These in fact have some of the lowest B-factors in the whole protein. The figure given (figure 1A) clearly does not show a zero mean. The authors should also discuss whether the two chains in the crystal structure were averaged. Furthermore, the equation is not used in the study cited.
Section 4.6: full parameters for the modelling should be given. Ideally, to make the modelling reproducible, the seed values used for each run should be provided.
Lines 315-323: as I have explained in my review, the conclusions of this manuscript are not supported by the data presented. A more realistic assessment of the data would be more convincing.
Comments on the Quality of English LanguageThe text of this manuscript is written in a manner that makes some sentences rather hard to understand. Unnecessary adjectives are a particular issue.
I have highlighted the most challenging parts to read in the comments to authors.
Reviewer 2 Report
Comments and Suggestions for Authors
In their work Xiang et al. described a method to increase the thermostability of a Thermomyces lanuginosus lipase (TLL) based on the B-factor of the crystal structure and multiple sequence alignment of other lipases. Their results suggest that this simple method may be of interest, because it allowed them to increase the thermostability of the mutants up to 5 °C. However, I have some additional comments and questions to their work:
- In the Introduction I’m missing a short review on other techniques for thermal stability enhancement which were applied to TLL.
- Why was only the crystal structure with PDB ID 4ZGB selected for the analysis? The PDB database offers more than one crystal structure of TLL with better resolution? Why did the authors pick only this structure and not the others? More structures would provide more information on the enzyme flexibility.
- Four of the five mutants showed enhanced thermostability – except for the D27N, which showed decreased residual activity and kinetics properties. Why is the substitution with Arginine more successful that with Asparagine in this case?
- To investigate the mechanism of thermostability the authors ran MD simulations of the wild type TLL and five mutants. I’m wondering if the authors saw any correlations between the B-factor of the selected residues with the RMSF values from MD simulations? How was the RMSF measured – for whole amino acids or only CA atoms?
- How was a salt bridge/ hydrogen bond defined? How was the probability calculated? Also, a list of the identified interactions between the 25-loop and N- and C-terminal residues would be helpful.
- Figure 6B is not very clear, in my opinion, as it should be showing the interactions between the N- and C-terminal regions of TLL and the 25-loop and no interactions are shown. Also, it is unclear whether it is a structure from MD simulations or starting point model. The same applies to Figure 7A – no interactions shown, and it’s unclear which structure is shown at the figure.
- I’m concerned that the length of the MD simulations was insufficient, as the systems were unable to equilibrate well. It is visible on the RMSD plot (Figure 8) – a plateau is visible around 15th ns of simulation time, so at about ¾ of the whole simulation. Which part of the MD did the authors analyse? ? In case of short simulations the flexibility/movement of residues may be caused by insufficient minimization/equilibration and not an indication of protein flexibility/rigidity. I would suggest extending the length of the MD simulations (at least up to 100 ns) and then analyse only the part in which the equilibrium (plateau on the RMSD plot) is reached.
- In the Discussion section the authors wrote that the mutations are not additive – do they know why is it happening? Does that mean that the 25-loop should preserve at least a small degree of flexibility in order to maintain its function? It would be interesting to see an MD simulation of the quadruple variants to measure the flexibility and h-bonds/salt bridges of the 25-loop.
Therefore I will recommend this manuscript to reconsider for publication after major revisions, mostly related to the description and analysis of the computational part of the work.
Round 2
Reviewer 2 Report
Comments and Suggestions for Authors
The authors have successfully addressed all my comments and suggestions.
However, some of their answers - regarding mostly the methodology of their analyses - can be easily paste into the manuscript to ensure reproducibility and, in some cases, clear the doubts of the reader.
Other than that - I do not have more suggestions nor comments.
